# Excreted *Trypanosoma brucei* proteins inhibit *Plasmodium* hepatic infection

**Adriana Temporão**[1⊙], **Margarida Sanches-Vaz**[1⊙], **Rafael Luís**[1], **Helena Nunes-Cabaço**[1], **Terry K. Smith**[2], **Miguel Prudêncio**[1]*, **Luisa M. Figueiredo**[1]*

**1** Instituto de Medicina Molecular João Lobo Antunes, Faculdade de Medicina, Universidade de Lisboa, Lisboa, Portugal, **2** Schools of Biology and Chemistry Biomedical Sciences Research Complex, The North Haugh, The University, St. Andrews, Scotland, United Kingdom

⊙ These authors contributed equally to this work.
* mprudencio@medicina.ulisboa.pt (MP); lmf@medicina.ulisboa.pt (LMF)

**Data Availability Statement:** All relevant data are within the manuscript and its Supporting Information files.

## Abstract

Malaria, a disease caused by *Plasmodium* parasites, remains a major threat to public health globally. It is the most common disease in patients with sleeping sickness, another parasitic illness, caused by *Trypanosoma brucei*. We have previously shown that a *T. brucei* infection impairs a secondary *P. berghei* liver infection and decreases malaria severity in mice. However, whether this effect requires an active trypanosome infection remained unknown. Here, we show that *Plasmodium* liver infection can also be inhibited by the serum of a mouse previously infected by *T. brucei* and by total protein lysates of this kinetoplastid. Biochemical characterisation showed that the anti-*Plasmodium* activity of the total *T. brucei* lysates depends on its protein fraction, but is independent of the abundant variant surface glycoprotein. Finally, we found that the protein(s) responsible for the inhibition of *Plasmodium* infection is/are present within a fraction of ~350 proteins that are excreted to the bloodstream of the host. We conclude that the defence mechanism developed by trypanosomes against *Plasmodium* relies on protein excretion. This study opens the door to the identification of novel antiplasmodial intervention strategies.

## Author summary

Malaria and sleeping sickness are parasitic illnesses that overlap geographically, making it likely that co-infections between the two causative parasites occur. It was previously shown that when mice are first infected with *Trypanosoma brucei*, there was an attenuation of the subsequent infection by *Plasmodium*. Here we sought to assess whether an active *T. brucei* infection was required for this impairment, and to unravel the mechanism behind this phenomenon. We found that not only *T. brucei* total lysates are able to inhibit *Plasmodium* liver infection, but also that mice that received these lysates are partly protected from developing severe malaria pathology. We further showed that this protective effect is mediated by proteins excreted by trypanosomes. Our study paves the way to the development of novel antiplasmodial intervention strategies, based on the mechanism involved during the co-infection between *T. brucei* and *Plasmodium*.

**Funding:** This work was supported by the Howard Hughes Medical Institute (ref. 55007419) awarded to LMF, by Fundação para a Ciência e Tecnologia (PD/ BD/138891/2018) awarded to AT, (PD/BD/ 105838/2018) awarded to MSV, (CEECIND/03322/ 2018) awarded to LMF, (CEECIND/03539/2017) awarded to MP, and by the Wellcome Trust (094476/Z/10/Z) awarded to TKS. The funders had no role in study design, data collection and analysis, decision to publish, or preparation of the manuscript.

**Competing interests:** The authors have declared that no competing interests exist.

## Introduction

The term co-infection describes the co-existence of two or more infectious agents in the same host [1], an event that occurs when that host is infected by different pathogens. This phenomenon can have a synergistic effect, when the infection by a microorganism facilitates the infection by another one; an antagonistic effect, when the presence of a microorganism inhibits a subsequent infection by a different one; or no effect, when the infection by a microorganism does not impact the infection by another. Co-infections have important repercussions on human and animal health because they can impact the host's susceptibility to other infectious agents, the duration of infection, the risk of transmission, the clinical symptoms, disease treatment and prevention strategies [2, 3].

*Plasmodium* is the causative agent of malaria, a disease that remains one of the major global public health threats worldwide. According to the World Health Organization (WHO), in 2019 there were 229 million cases of malaria and 409 000 deaths were caused by this disease [4]. Humans become infected once an infected female *Anopheles* mosquito takes a blood meal, injecting sporozoites into the host's skin. These sporozoites reach the blood circulation and travel to the liver, where they cross endothelium sinusoids and traverse several liver cells until they effectively infect one. Once in the hepatocyte, sporozoites differentiate into exoerythrocytic forms (EEFs) that ultimately originate thousands of merozoites. These newly formed parasites egress from the hepatocytes and infect red blood cells, thereby initiating the blood stage of the *Plasmodium* infection [5]. Until recently, the liver stage of *Plasmodium* infection was largely overlooked and its asymptomatic nature led to the widespread assumption that hepatic parasites were undetected by the immune system [6]. Nevertheless, this notion has now been disproven, and multiple studies have shown that an immune response is mounted in this stage of the infection [7, 8]. The process of antigen presentation is initiated as soon as the *Plasmodium* sporozoites start traversing and invading hepatocytes [9]. During this phase, several liver cells, including Kupffer cells, are exposed to parasite antigens and may present them to lymphocytes [10, 11].

*Trypanosoma brucei*, the causative agent of human African trypanosomiasis (HAT) or sleeping sickness, is a protozoan parasite that shares its geographical distribution in the sub-Saharan African region with *Plasmodium*. *T. brucei* parasites are transmitted by an infected *Glossina* fly upon a blood meal from a mammalian host. Parasites released into the skin during this process rapidly differentiate into proliferative forms that invade the bloodstream and several tissues, including the brain, adipose tissue and skin [12, 13]. Conversely, when a *Glossina* fly bites an infected mammal, it ingests parasites that differentiate into proliferative procyclic forms in the tsetse midgut [14]. The immune response against *T. brucei* relies mainly on the activation of lymphocytes and macrophages, which subsequently produce an array of cytokines [15, 16]. Numerous parasite factors, such as proteins, trigger several immune signalling pathways [17]. The variant surface glycoprotein (VSG) is the most abundant protein in the parasite's surface membrane and is tightly associated with the activation of both T and B lymphocytes, as well as of macrophages [18].

There are several reports of co-infections between *Plasmodium* and bacteria [19, 20], viruses [21, 22] and even other parasites [23–25]. Given that human infections by *Plasmodium* and *T. brucei* largely overlap geographically in Africa, co-infections with these two pathogens are likely to occur. In fact, malaria is commonly found in patients with HAT [26]. Recently, it was shown that an ongoing infection by *T. brucei* strongly impairs a secondary infection by *P. berghei* sporozoites in mice [27], similarly to what has been previously shown for the co-infection between *Schistosoma* and *Plasmodium* [28].

A total of 254 *T. brucei* proteins have been identified in the plasma of *T. b. rhodesiense*-infected patients with late-stage sleeping sickness [29]. The high number of proteins found may be due to the several waves of parasitaemia, that allow for the accumulation of

trypanosome-derived proteins in the bloodstream of the patients. These proteins may be secreted or shed from live parasites or even the result of parasite lysis triggered by the host immune system. Although VSG is the most abundant of these proteins, this finding highlights the existence of an array of other proteins, which may interact with the host [29]. A proteomic analysis of *T. brucei* excretome/secretome not only identified 444 proteins, unveiling new potential therapeutic targets or diagnostic markers, but also suggested that some proteins may not employ the classical secretory pathway and may also have secondary roles, in addition to their canonical one, once they are excreted/secreted [30].

Having previously shown that a *T. brucei* infection is able to impair a secondary *P. berghei* liver infection in mice [27], we sought to assess whether an active *T. brucei* infection was required for this impairment, and to unravel the mechanism behind this phenomenon. We observed that *Plasmodium* liver infection is markedly inhibited by *T. brucei* total lysates, which partly protect animals from developing severe malaria pathology. Further characterisation of these lysates revealed that proteins excreted by trypanosomes are able to inhibit *Plasmodium* hepatic infection. These results pave the way to the investigation of a novel mechanism at play during a co-infection between two different parasite species, which may lead to the development of novel antiplasmodial intervention strategies.

## Materials and methods

### Ethics statement

All the experimental animal work was performed in strict compliance to the guidelines of our institution's animal ethics committee, who also approved this study (under authorization AWB_2015_09_MP_Malaria), and in accordance with the Federation of European Laboratory Animal Science Associations (FELASA) guidelines.

### Experimental animals

Male C57BL/6J mice (6–8 weeks old) were purchased from Charles River Laboratories (Lyon, France). *RAG2*$^{-/-}$*γc*$^{-/-}$ mice, kindly provided by the Instituto de Medicina Molecular João Lobo Antunes (iMM JLA, Lisbon, Portugal)'s Silva-Santos laboratory, were bred in specific-pathogen-free (SPF) facilities at iMM JLA. All animals were housed and kept in the SPF rodent facility of the iMM JLA, a licensed establishment that complies with the European Directive 2010/63/EU on the protection of animals used for scientific purposes.

### Parasites and infections

The transgenic 90–13 cell-line of the *Trypanosoma brucei brucei* AnTaT 1.1 pleomorphic strain [31], as well as the monomorphic *Lister* 427 strain [32] (antigenic type MiTat 1.2, clone 221a, that co-expresses the T7 RNA polymerase and the tetracyclin repressor genes), were employed throughout this study. Bloodstream forms (BSF) were grown in HMI-11 medium at 37°C and 5% $CO_2$ [33]. Procyclic forms (PCF), obtained by differentiation with *cis*-aconitate [34] of the monomorphic *Lister* 427 strain, were grown in Differentiating Trypanosome Medium (DTM) at 27°C. BSF were employed in all the experiments, unless stated otherwise. Mice were infected by intraperitoneal (i.p.) injection of 2 x $10^3$ motile bloodstream parasites, obtained from cryostabilates of the pleomorphic strain, as previously described [12, 35].

GFP-[36] and Luciferase-[37]expressing *Plasmodium berghei* ANKA and *Plasmodium yoelii yoelii* 17XNL rodent malaria parasites were used throughout this study. For intravenous (i.v.) sporozoite injections, *P. berghei* and *P. yoelii* sporozoites were obtained by dissection of salivary glands from infected female *Anopheles stephensi* mosquitoes, reared at iMM JLA. Mice

were infected by retro-orbital i.v. injection of 500 or 3 x $10^4$ sporozoites, under isoflurane anaesthesia.

## Assessment of Experimental Cerebral Malaria (ECM) symptoms

ECM development was monitored daily using the rapid murine coma and behaviour scale (RMCBS) score, as described in [38]. Mice with RMCBS score equal or below 5/20 were classified as displaying ECM symptoms and euthanized immediately.

## Assessment of *Plasmodium* parasitaemia

*P. berghei* parasitaemia was assessed using a modified version of the protocol described in Zuzarte-Luis V. *et al* [39]. Briefly, 5 μl of blood was collected from the tail vein into 45 μl of lysis buffer and preserved at -20˚C. Parasite bioluminescence was measured by adding 50 μl of D-Luciferin dissolved in luciferase assay buffer to 15 μl of the blood lysate. Luminescence was immediately measured on a microplate reader (Tecan M200, Switzerland).

## Quantification of *Plasmodium* hepatic infection *in vivo*

Mouse liver infection by *Plasmodium* was assessed 46 h after sporozoite inoculation and quantified by quantitative real-time reverse transcriptase-PCR (qRT-PCR), as previously described [40]. For qRT-PCR analyses, 0.7–0.9 mg of livers collected upon euthanasia of infected mice were mechanically homogenized in TRIzol (BioRad, Hercules, CA, USA), RNA was extracted following the manufacturer's instructions, and converted into complementary DNA (cDNA) as described below. Liver *Plasmodium* load was quantified by qRT-PCR, as previously described[41], using primers specific for *Plasmodium* 18S rRNA (**Table 1**). Expression of the endogenous mouse housekeeping gene hypoxanthine-guanine phosphoribosyltransferase (*Hprt*) was used for normalization.

## RNA extraction, complementary DNA synthesis and qRT-PCR

RNA was extracted following the manufacturer's instructions (BioRad, Hercules, CA, USA) and quantified using a NanoDrop DR 2000 Spectrophotometer (Thermo Fisher Scientific, Waltham, MA USA). cDNA was synthesized from 1 μg of RNA, using the NZYTech cDNA synthesis kit (NZYTech, Lisbon, Portugal), according to the manufacturer's recommendations, and employing the following thermocycling parameters: 25˚C for 10 min, 55˚C for 30 min, and 85˚C for 5 min. The qRT-PCR reaction was performed in a total volume of 10 μl employing an QuantStudio 5 equipment (Applied Biosystems, Foster City, CA, USA), using the SYBR Green PCR Master Mix (Applied Biosystems, Foster City, CA, USA) and the following thermocycling parameters: 50˚C for 2 min, 95˚C for 10 min, 40 cycles at 95˚C for 15 s and 60˚C for 1 min; melting stage employed 95˚C for 15 s, 60˚C for 1 min, and 95˚C for 30 s. Primer pairs used to detect target gene transcripts are listed in **Table 1**. Gene expression was analysed

**Table 1. List of primer sequences used for qRT-PCR.**

| Target gene | Forward primer | Reverse primer |
| --- | --- | --- |
| *P. berghei* 18S rRNA | AAGCATTAAATAAAGCGAATACATCCTTAC | GGAGATTGGTTTTGACGTTTATGTG |
| Hprt | TTTGCTGACCTGCTG GATTAC | CAAGACATTCTTTCCAGTTAAAGTTG |
| CLEC4f | TGAGTGGAATAAAGAGCCTCCC | TCATAGTCCCTAAGCCTCTGGA |
| CD68 | AGCTGCCTGACAAGGGACACT | AGGAGGACCAGGCCAATGAT |
| F4/80 | CCCAGCTTATGCCACCTGCA | TCCAGGCCCTGGAACATTGG |

by the comparative CT ($\Delta\Delta$CT) method and the expression level of all target genes was normalized to that of *Hprt*.

## Serum transfer

Blood was collected by heart puncture of either naïve mice or mice infected 5 days earlier with $2 \times 10^3$ *T. brucei* parasites. After clotting, samples were centrifuged at 1,000 x g for 10 min at room temperature (RT). Serum was transferred to a fresh tube and pooled together. 200 or 450 μl were administered into naïve mice by i.v. injection in the tail vein, 30 min prior to *P. berghei* sporozoite inoculation.

## *T. brucei* lysate preparation

*T. brucei* lysates were obtained from the monomorphic *Lister* 427 strain. Trypanosomes were harvested by centrifugation at 3,000 x g for 15 min at 4˚C, washed in phosphate-buffered saline (PBS), followed by centrifugation at 3,000 x g for 15 min at 4˚C and resuspension in PBS. Trypanosomes were kept on ice and mechanically disrupted, using the Soniprep 150 (MSE, London, UK), following a sequence of 6 cycles of 1 min 30 s sonication at maximum intensity and 30 s pauses. Parasite lysis was confirmed by microscopy, flash frozen in liquid nitrogen and stored at -80˚C. *T. brucei* lysates were inoculated into mice through i.v. injection in the tail vein. Livers were collected from mice 46 h after sporozoite injection and processed as described above for the quantification of *Plasmodium* liver load by qRT-PCR.

## Immunofluorescence microscopy analysis of *in vivo P. berghei* hepatic infection

Livers were collected at 46 h after *P. berghei* sporozoite inoculation and fixed in 4% (v/v) paraformaldehyde (PFA) (SantaCruz Biotechnology, Dallas, TX, USA) for at least 12 h at RT. Liver sections of 50 μm thickness were stained and analyzed as previously described [40, 42]. Briefly, slices were incubated in permeabilization/blocking solution containing 1% (w/v) bovine serum albumin (Sigma-Aldrich, St. Louis, MO, USA) and 0.5% (v/v) Triton-X100 in PBS (Sigma-Aldrich) at RT for 1 h, followed by a 2 h incubation at RT with an anti-UIS4 antibody (goat, homemade; dilution 1:500). Liver sections were further incubated for 1 hour in a 1:500 dilution of anti-GFP-Alexa 488 antibody (Invitrogen, Carlsbad, CA, USA) and anti-goat Alexa-Fluor 568 (Invitrogen, Carlsbad, CA, USA) in the presence of a 1:1,000 dilution of Hoechst 33342 (Invitrogen, Carlsbad, CA, USA). After washing, liver sections were mounted on microscope slides with Fluoromount (SouthernBiotech, Birmingham, AL, USA). Widefield images for the determination of the size of *P. berghei* intrahepatic forms were acquired employing a Zeiss Axiovert 200M microscope (Carl Zeiss, Oberkochen, Germany). Confocal images were acquired using a Zeiss LSM 510 confocal microscope (Carl Zeiss, Oberkochen, Germany). Images were processed with the ImageJ software (version 1.47, NIH, Bethesda, MD, USA).

## Liver histopathology

Livers collected 46 h after *P. berghei* sporozoite inoculation were formalin-fixed in neutral buffered formalin, paraffin-embedded, cut in 4 μm sections, and stained with hematoxylin (Bio-Optica, Milan, Italy) and eosin (Thermo Fisher Scientific, Waltham, MA USA). Tissue sections were analyzed by a pathologist blinded to experimental groups, using a Leica DM2000 microscope coupled to a Leica MC170 HD microscope camera (Leica Microsystems, Wetzlar, Germany).

## Biochemical assessment of liver pathology

Blood was collected from mice either naïve, infected 5 days earlier with *T. brucei*, or injected 30 min earlier with *T. brucei* lysates. Blood was collected at the time of *P. berghei* sporozoite injection, 12 h and 24 h later. After clotting, samples were centrifuged at 1,000 x g for 10 min at 4˚C, and serum was transferred to a fresh tube. Liver pathology was assessed by quantification of serum levels of alanine transaminase (ALT), aspartate transaminase (AST), bilirubin (total) and glucose (Idexx Bioresearch, Stuttgart, Germany).

## *In vivo* depletion of macrophages

Macrophages were depleted by i.v. injection of 1 mg of liposome-encapsulated clodronate (Liposoma B.V., Amsterdam, The Netherlands), 48 h prior to *P. berghei* sporozoite inoculation. An equivalent volume of liposome-encapsulated PBS was injected i.v. into control mice. Livers were collected 46 h after sporozoite injection and processed as described above for the quantification of *P. berghei* liver load by qRT-PCR.

## *In vivo* inhibition of monocyte migration

Monocyte recruitment was inhibited by i.p. injection of 20 μg anti-CCR2 antibody, kindly provided by Matthias Mack (University Hospital Regensburg), 48 and 24 h prior to inoculation of *P. berghei* sporozoites, at the time of infection and 24 h later. An equivalent volume of PBS was injected i.p. into control mice. Livers were collected 46 h after sporozoite injection and processed as described above for the quantification of the *P. berghei* liver load by qRT-PCR.

## Isolation and quantification of liver monocytes by flow cytometry

Livers were collected by dissection of mice at 46 hours after *P. berghei* sporozoite inoculation, mechanically homogenized in 5 ml of PBS containing 2 U/ml DNAse (Sigma-Aldrich, St. Louis, MO, USA), using a 100 μm cell strainer, and centrifuged at 410 x g for 8 min at RT. The cell pellet was fractionated using 10 ml of 35% (v/v) of Percoll gradient medium diluted in non-supplemented RPMI (Gibco-Thermo Fisher Scientific, Waltham, MA USA), followed by centrifugation at 1,360 x g for 20 min without brake at 20˚C. The cells deposited in the pellet were washed with PBS and centrifuged at 410 x g for 8 min at RT. The cell pellet was depleted of red blood cells (RBCs) by incubation in 3 ml of ammonium-chloride-potassium solution, following incubation for 3 min at RT and subsequent inactivation with 7 ml of PBS 2% FBS (FACS buffer). The leukocyte suspension was centrifuged at 410 x g for 8 min at RT and resuspended in 500 μl of PBS for subsequent staining. One million leukocytes from each mouse were then plated in 96-well plates, centrifuged at 845 x g for 3 min at 4˚C and incubated with α-CD16/CD32 (eBioscience/Thermo Fisher Scientific, Waltham, MA, USA) for 20 min at 4˚C. For the staining of lymphoid surface antigens, cells were incubated for 20 min at 4˚C with FACS buffer containing Fixable Viability Dye (eFluor 780) and the following conjugated flow cytometry monoclonal antibodies: FITC anti-CD11b (clone M1/70), BV605-Ly6C (clone HK1.4), BV785-MHC II (clone M5/114.15.2), PE-Cy7-Ly6G (clone 1A8) and PE-CD11c (clone N418). All antibodies were from eBioscience (Thermo Fisher Scientific, St. Louis, MO, USA) or Biolegend. Cells were acquired on a FACS LSRFortessa X-20 (BD Biosciences Franklin Lakes, New Jersey, U.S.) and data acquisition and analysis were carried out using the FACSDiva (version 6.2) and FlowJo (version 10.7, FlowJo) software packages, respectively.

### *In vivo* hematopoietic cell ablation

Lymphoid and myeloid cell ablation was obtained by submitting mice to whole-body γ-radiation, using the Gammacell 3000 ELAN irradiator (Elan Drug Technologies, Dublin, Ireland), at a dose of 9 Gy for 2 min at room temperature, 24 h prior to *P. berghei* sporozoite inoculation. Livers were collected 46 h after sporozoite injection and processed as described above for the quantification of the *P. berghei* liver load by qRT-PCR.

### *T. brucei* Lysate digestion with DNAse

DNA present in the lysates was digested in PBS containing 20 U/ml of DNAse (Sigma-Aldrich, St. Louis, MO, USA) and 10% (v/v) of enzyme buffer, following incubation for 2 h at 37°C and subsequent DNAse inactivation at 75°C for 10 min. Untreated lysates were submitted to the same incubation periods and temperatures. Treated or untreated lysates of $1 \times 10^8$ trypanosomes (200 μl) were injected i.v. into mice 30 min prior to *P. berghei* sporozoite inoculation. An equivalent volume of PBS containing inactivated DNAse was injected i.v. into control mice.

### *T. brucei* Lysate digestion with proteinase K

Proteins present in the lysates were digested in PBS containing 250 μg/ml of proteinase K (Promega, Wisconsin, USA) and 30% of digestion buffer (0.01 M Tris, 5 mM ethylenediamine tetraacetic acid [EDTA], 0.5% sodium dodecyl sulphate [SDS]), following overnight incubation at 50°C and subsequent proteinase K inactivation at 95°C for 10 min. Untreated lysates were submitted to the same incubation periods and temperature. Treated or untreated lysates of $10^8$ trypanosomes (200 μl) were injected i.v. into mice 30 min prior to *P. berghei* sporozoite inoculation. An equivalent volume of PBS containing inactivated proteinase K was injected i.v. into control mice.

### Trypanosome digestion with phospholipase C (PLC)

Trypanosomes were harvested and pelleted as described above and suspended in PBS containing 0.5 U/ml of PLC from *C. perfringens* (Sigma-Aldrich, St. Louis, MO, USA) at a concentration of $1 \times 10^8$ trypanosomes/100 μl. The parasite suspension was incubated for 30 min at 37°C and then centrifuged at 10,000 x g for 30 s, cleaving and separating the glycosylphosphatidylinositol (GPI)-anchored proteins (supernatant) from the residual intact parasites (pellet). The parasite pellet was suspended in PBS. The PLC present in the soluble GPI-anchored proteins suspension was inactivated at 95°C for 5 min. The equivalent to $10^8$ trypanosomes (200 μl) was injected i.v. into mice 30 min prior to *P. berghei* sporozoite inoculation. An equivalent volume of PBS containing inactivated PLC was injected i.v. into control mice.

### sVSG purification *T. brucei* lysates

Soluble VSG (sVSG) was obtained through cleavage of membrane form VSG by the endogenous GPI-PLC. *T. brucei* were grown as described above, transferred to Falcon tubes and incubated on ice for 10 min. The parasites were then centrifuged at 2,500 x g, 0°C, 10 min, washed with trypanosome dilution buffer (20 mM $Na_2HPO_4$, 2 mM $NaH_2PO_4$, 80 mM NaCl, 5 mM KCl, 1 mM $MgSO_4$, 20 mM glucose; pH adjusted to 7.7) and resuspended in a, pre-warmed at 37°C, 10 mM sodium phosphate buffer (pH 8.0) containing 2 mM phenylmethylsulfonyl fluoride (PMSF) (Sigma-Aldrich) and 1:200 Protease Inhibitor Cocktail (PIC) (Sigma-Aldrich), followed by incubation at 37°C for 5 min. The parasite suspension was then centrifuged at maximum speed for 5 min, the supernatant was collected and passed through a column of

DE52, pre-equilibrated in 10 mM sodium phosphate (pH 8.0). The eluate containing the sVSG was washed with Milli-Q $H_2O$ (Millipore, Massachusetts, USA) and concentrated using an Amicon Ultra-15 3 kDa (Millipore, Massachusetts, USA). The purity of sVSG was analysed by SDS-PAGE stained with Coomassie Blue. 100 μg of sVSG, equivalent of $1 \times 10^8$ trypanosomes, was injected i.v. into mice 30 min prior to *P. berghei* sporozoite inoculation. An equivalent amount of albumin (Sigma-Aldrich, St. Louis, MO, USA) in PBS was injected i.v. into control mice. Livers were collected 46 h after sporozoite injection and processed as described above for the quantification of *P. berghei* liver load by qRT-PCR.

## Excretion/Secretion of *T. brucei* proteins

Trypanosomes were harvested and pelleted as described above and incubated in PBS for 1 h at RT at a concentration of $10^8$ trypanosomes/200 μl. The parasite suspension was then centrifuged at 10,000 x g for 30 s, separating excreted/secreted proteins (supernatant) from the parasites (pellet). The equivalent to $10^8$ trypanosomes (200 μl) was injected i.v. in the tail vein of mice 30 min prior to inoculation of *P. berghei* sporozoites.

## Extracellular vesicle purification

Trypanosomes were harvested and pelleted as described above. The supernatant was filtered (0.2 μm) to remove remaining parasites and cell debris. Extracellular vesicles (EVs) were collected by ultracentrifugation at 100,000 x g for 70 min, as described in [43]. EVs were stored in PBS at 4°C. Each mouse was injected with EVs excreted by $10^8$ trypanosomes.

## Transmission electron microscopy

EVs were incubated on a glow-discharged, carbon-formvar 100-mesh TEM grid and fixed in 4% formaldehyde in 0.2M Phosphate buffer for 5 min. Samples were washed with dH2O and incubated for 5 min with 2% aqueous uranyl acetate. TEM micrographs were acquired on a Tecnai G2 Spirit BioTWIN TEM (FEI) operated at 120 kV using a side mount 2k x 2k Olympus-SIS Veleta CCD camera.

## Mass spectrometry analysis

Protein mass spectrometry was carried out by the inhouse facility at the University of St. Andrews. Samples were reduced with DTT, alkylated with iodoacetamide and digested at 37°C with trypsin. Samples in gels were extracted from diced gel pieces in 5% formic acid to an extraction volume of 20 μl. A proportion of the sample was injected onto an eksigent nano-LC set up in 'trap elute' configuration using a Pepmap column and trap (ThermoScientific). Peptides were eluted in a linear gradient over 180 min and flowed directly into a Sciex 5600 + Q-Tof mass spectrometer. Survey scans were carried out between 400–1200 m/z and the strongest 15 peptides from each scan were fragmented to give MS-MS spectra from 100–2000 m/z. Spectra were extracted using the mgf generator script from Sciex and the data searched using the mascot search algorithm against *T. brucei* strain 427 genome and NCBI databases.

## Gene ontology analysis

Gene ontology (GO) categories related to biological processes, molecular function and cell component were identified using TopGo, based on the annotated GO term list for *T. brucei* TREU927 genome version 2.1 available in TritrypDB [44]. Enriched GO terms among detected proteins were established using the weight01 algorithm and Fisher's exact test at a false discovery rate (FDR) p-value $\leq 0.05$.

### Statistical analyses

Data are expressed as mean ± standard error of the mean (SEM). Statistical analyses were performed using the GraphPad Prism 6 software (La Jolla, CA, USA). Statistically significant differences were determined using One-way analysis of variance (ANOVA), Log- Rank (Mantel-Cox) test, Two-tailed Mann-Whitney test or Unpaired t test. Statistical significances are represented as indicated in each figure: ns–not significant, $*$ $p < 0.05$, $**$ $p < 0.01$, $***$ $p < 0.001$ and $****$ $p < 0.0001$.

## Results

### Total lysates of *T. brucei* protect mice against *P. berghei* liver infection

Having previously shown that an ongoing infection by *T. brucei* impairs a secondary *P. berghei* hepatic infection [27], we initially assessed whether the observed protection could be transferred to a naïve mouse through passive transfer of the serum of a *T. brucei*-infected animal. To this end, 200 and 450 µl of serum from donor mice, either naïve or infected 5 days earlier with *T. brucei*, were collected and injected i.v. into naïve recipient mice. Thirty min later, the recipient mice were inoculated with *P. berghei* sporozoites and its hepatic infection was assessed by qRT-PCR 46 h later. We found that passive transfer of serum from *T. brucei*-infected animals into naïve mice led to a dose-dependent impairment of *P. berghei* hepatic infection relative to those that received serum from healthy animals (**Fig 1A**).

Knowing that the serum contains components from both the host and the parasites, we sought to assess whether the inhibitory component of the serum was trypanosome-derived. To this end, total lysate suspensions of $10^6$, $10^7$ and $10^8$ mechanically lysed trypanosomes were injected intravenously into mice 30 min prior to inoculation of 3 x $10^4$ *P. berghei* sporozoites (**Fig 1B**). In parallel, equivalent suspensions of total lysates of *P. berghei* blood stages were injected into mice (**Fig 1B**), to test the specificity of the effect of the trypanosome lysates, and to control for the injection of high doses of parasite-derived molecules. Our results show that *T. brucei* total lysates are able to impair a *P. berghei* liver infection in a dose-dependent manner, with the suspension containing $10^8$ trypanosomes showing the highest level of protection (~80%) (**Fig 1C**). This effect appears to be trypanosome-specific, since the control mice that received the *P. berghei* blood stage lysates displayed a hepatic infection similar to that of animals infected with *Plasmodium* only. Additionally, we assessed the capacity of trypanosome total lysates to impair hepatic infection by another *Plasmodium* species, *P. yoelii*. Our results show that *T. brucei* infection and *T. brucei* lysates had a similar ~76% inhibitory effect on *P. yoelii* liver infection (**S1 Fig**). Altogether, our results show that *T. brucei* total lysates administered to mice prior to sporozoite inoculation significantly reduce a subsequent *Plasmodium* hepatic infection. Moreover, our data suggest that the inhibitory effect is trypanosome-specific and that it is not limited to a single *Plasmodium* species.

Having observed that the administration of *T. brucei* total lysates leads to a significant decrease in a subsequent *P. berghei* hepatic infection, we next investigated whether this effect was due to a reduction in the number of *P. berghei*-infected hepatocytes or to a defect in the parasite's ability to replicate inside the hepatocytes. To this end, we analysed sections of livers collected from mice infected only with *P. berghei*, co-infected with both parasites, or injected with the *T. brucei* total lysates prior to *P. berghei* sporozoite inoculation, by immunofluorescence microscopy. We observed that both the co-infected mice and those that received the *T. brucei* total lysates show a ~75% reduction in the number of infected hepatocytes, when compared to the control mice infected only with *P. berghei* (**Fig 1D**). Conversely, no significant differences were observed in the size of the *P. berghei* EEFs in the three groups of mice

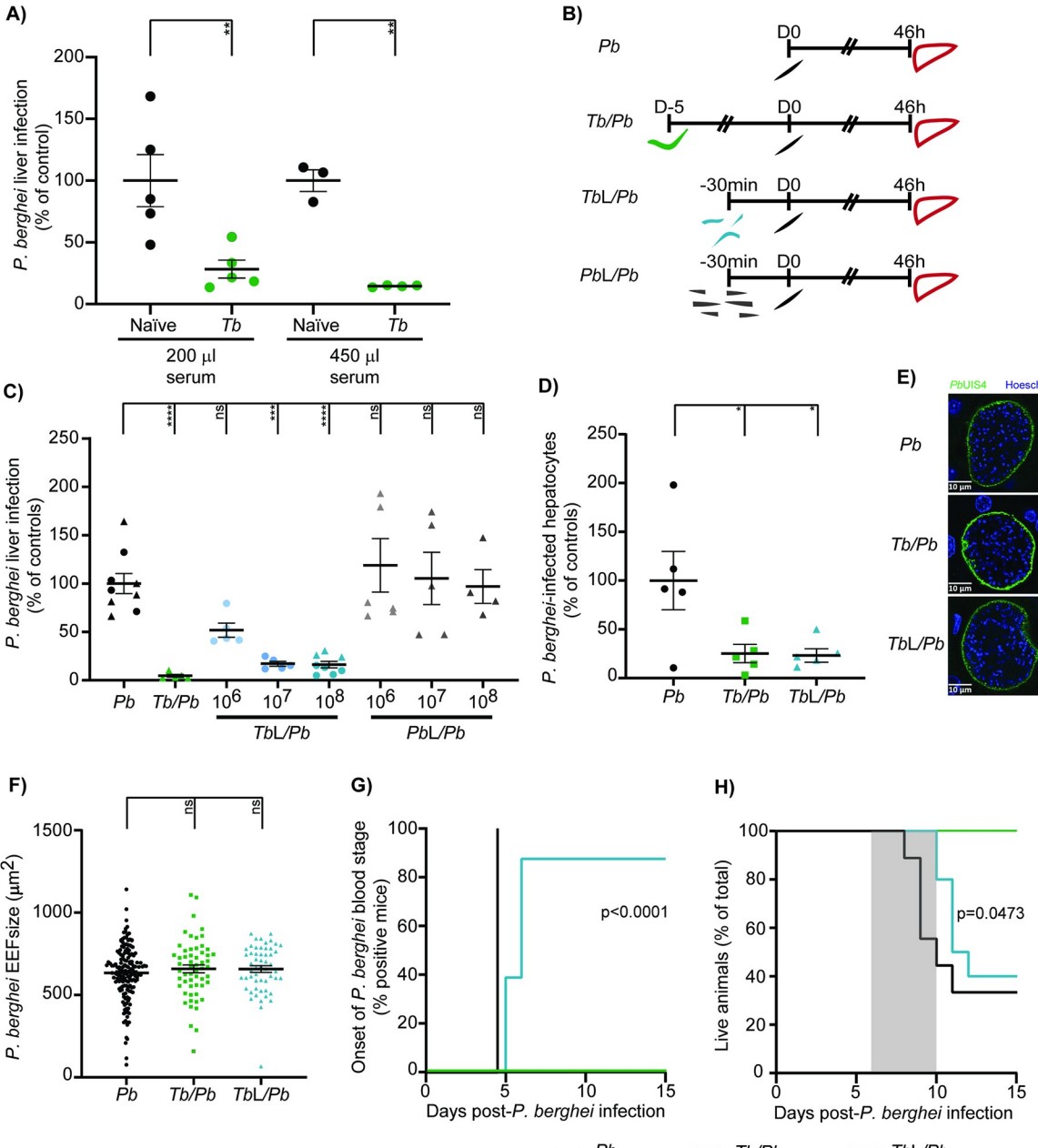

**Fig 1. Total lysates of *T. brucei* protect mice against *P. berghei* liver infection. (A)** *P. berghei* liver infection load quantified by qRT-PCR 46 h after injection of *P. berghei* sporozoites into mice that received 200 or 450 μl of serum from naïve mice (black symbols), or from mice infected 5 days earlier with *T. brucei* (green symbols). Symbols represent the individual values of each mouse in one independent experiment and error bars indicate the SEM. **(B)** Schematic illustration of the experimental design: (i) *P. berghei* single infection consists of an i.v. injection of sporozoites (*Pb*—black) and collection of livers 46 h later, (ii) Co-infection consists of infection with *T. brucei* parasites (*Tb/Pb*—green) by i.p. injection 5 days prior to sporozoite inoculation, and liver collection 46 h later (iii) *T. brucei* (*Tb*L/*Pb*—blue) and *P. berghei* (*Pb*L—grey) total lysates injection consists of an i.v. injection of the respective lysates 30 min prior to sporozoite inoculation and subsequent liver collection 46 h later. **(C)** Liver load was determined by qRT-PCR 46 h post-sporozoite inoculation in mice infected with *P. berghei* (*Pb*—black symbols), mice infected with *T. brucei* 5 days prior to sporozoite inoculation (*Tb/Pb*—green symbols; 2 independent experiments), mice previously injected with three different amounts of *T. brucei* total lysates (*Tb*L/*Pb*—blue symbols; 2 independent experiments) or mice injected with three different amounts of blood stage *P. berghei* total lysates (grey symbols; 1 independent experiment). Symbols represent the *Plasmodium* parasite load of individual mice. Error bars indicate the SEM. **(D)** Number of *P. berghei*-infected hepatocytes per square millimetre of liver section quantified by immunofluorescence microscopy 46 h after injection of *P. berghei* sporozoites into naïve mice (*Pb*—black symbols), mice infected 5 days earlier with *T. brucei* (*Tb/Pb*—green symbols), or that received *T. brucei* total lysates prior to *Plasmodium* sporozoites (*Tb*L/*Pb*—blue symbols). Symbols represent the values of individual mice from one independent experiment and error bars indicate the SEM. **(E)** Representative confocal microscopy

images of EEFs 46 h after injection of *P. berghei* sporozoites into naïve mice, mice infected 5 days earlier with *T. brucei* or mice that received *T. brucei* total lysates. Blue: Hoechst—nuclear; staining; green: *P. berghei* GFP labelling showing the parasite UIS4 protein. Scale bars, 10 μm **(F)** EEF area assessed by immunofluorescence microscopy, at 46 h after injection of *P. berghei* sporozoites into naïve mice (*Pb* -black symbols), mice infected 5 days earlier with *T. brucei* (*Tb/Pb*—green symbols), or mice that received *T. brucei* total lysates (*TbL/Pb*—blue symbols). Results are expressed as mean values of one experiment and error bars indicate the SEM. **(G)** Assessment of *P. berghei* prepatency period following inoculation of *P. berghei* sporozoites into naïve mice (*Pb*—black line), mice infected 5 days earlier with *T. brucei* (*Tb/Pb*—green line), or mice that previously received *T. brucei* total lysates (*TbL/Pb*—blue line). Percentage of mice displaying *P. berghei* parasitaemia, as measured by bioluminescence. The pooled data of 10 mice from two independent experiments is shown. **(H)** Mouse survival following inoculation of *P. berghei* sporozoites into naïve mice (*Pb*—black line), mice infected 5 days earlier with *T. brucei* (*Tb/Pb*—green line), or mice that received *T. brucei* total lysates (*TbL/Pb*—blue line). Percentage of live mice from a pool of 10 mice employed in two independent experiments. For **(C)**, **(D)** and **(E)** the one-way ANOVA with post-test Dunnett was employed to assess the statistical significance of differences between experimental groups. ns, not significant, $^*p<0.05$, $^{**}p<0.01$, $^{***}p<0.001$ and $^{****}p<0.0001$. For **(G)** and **(H)** the Mantel-Cox (log rank) test was employed to compare the onset of *P. berghei* parasitaemia.

(**Fig 1E and 1F**). Collectively, these data indicate that the impairment of *P. berghei* liver infection by *T. brucei* total lysates results from a decrease in the number of infected hepatocytes rather than from an impairment of the intrahepatic replication of *Plasmodium* parasites.

Next, we investigated whether the reduction in liver infection impacted the ensuing *Plasmodium* blood stage of infection [27]. To this end, animals were infected by i.v. injection of 500 luciferase-expressing *P. berghei* sporozoites in order to mimic the inoculum delivered by the bites of five infected mosquitoes [5]. As expected based on previous observations, 100% of the co-infected mice used as controls in this experiment were protected against *Plasmodium* erythrocytic infection (**Fig 1G**) [27]. Conversely, 90% of the mice that received the *T. brucei* total lysates developed an erythrocytic infection. Crucially, however, the prepatent period in these mice was one to two days longer than in mice infected only with *P. berghei*, indicating that, as expected, the reduction in the *P. berghei* liver burden in the mice receiving *T. brucei* total lysates leads to a delay in the onset of *Plasmodium* parasitaemia [39] (**Fig 1G**). Also of note, whereas 60% of *P. berghei* single infected mice died within 8 to 10 days after sporozoite inoculation with signs of experimental cerebral malaria (ECM), only 20% of the mice that received the *T. brucei* total lysates displayed ECM symptoms or succumbed within that time period (**Fig 1H**).

Overall, our data show that injection of *T. brucei* total lysates prior to sporozoite inoculation enhances resistance to *P. berghei* infection. *T. brucei* total lysates lead not only to a reduction in *P. berghei* liver burden, but also to a delay in the appearance of *Plasmodium* parasitaemia and protection against the development of severe malaria pathology.

## The protective effect of *T. brucei* lysates is time-dependent

Knowing that *P. berghei* hepatic infection can be impaired by *T. brucei* total lysates in the absence of an active infection, we next investigated the duration of this inhibitory effect following injection of trypanosome lysates into mice. To achieve this, *T. brucei* total lysates were injected into mice 48 h, 24 h, 12 h, 6 h and 30 min prior to sporozoite inoculation, and *P. berghei* liver infection was assessed 46 h after sporozoite injection (**Fig 2A**). The strongest reduction of *P. berghei* hepatic infection was observed when the *T. brucei* total lysates were injected 30 min prior to sporozoite inoculation, and the approximate half maximal inhibitory effect was observed when sporozoites were injected 12 h after lysate administration (**Fig 2B**). These results show that the protective effect of *T. brucei* total lysates is time-dependent, since this protection decreases as the period between the injection of the trypanosome lysates and the sporozoites increases.

Having shown the impact of the injection of *T. brucei* total lysates prior to *P. berghei* sporozoite inoculation, we then wondered whether the trypanosome lysates would impact a pre-established *P. berghei* liver infection. To evaluate this, *T. brucei* total lysates were injected into

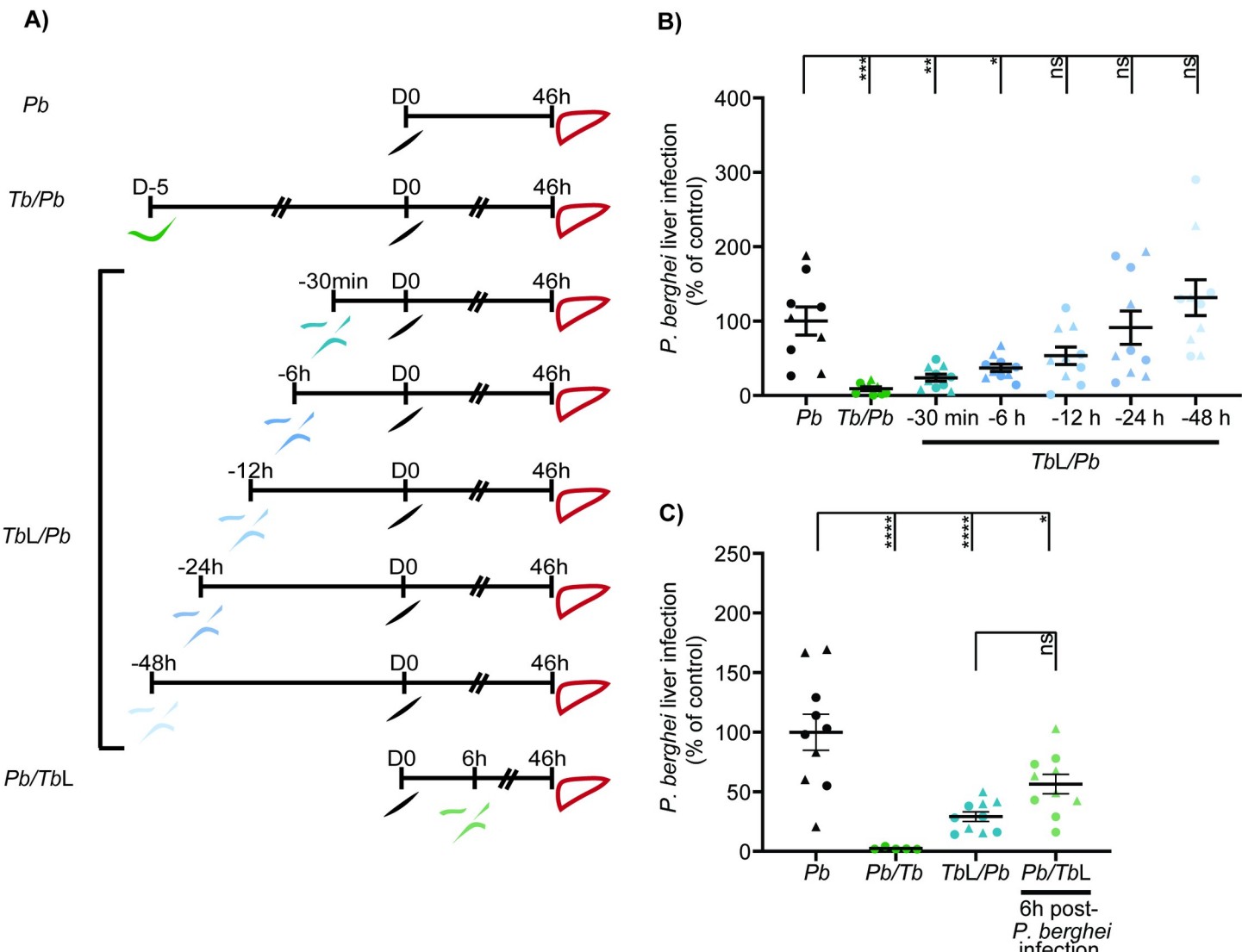

**Fig 2. The protective effect of _T. brucei_ lysates is time dependent. (A)** Schematic illustration of the experimental design: (i) _P. berghei_ single infection group consists of an i.v. injection of sporozoites (_Pb_—black) and collection of livers 46 h later, (ii) Co-infection consists of infection with _T. brucei_ parasites (_Tb/Pb_—green) by i.p. injection 5 days prior to sporozoite inoculation, and liver collection 46 h later (iii) _T. brucei_ total lysates group consists of an i.v. injection 30 min, 6 h, 12 h, 24 h, and 48 h prior to sporozoite inoculation (_TbL/Pb_—blue) and 6 h post sporozoite inoculation (light green) and subsequent liver collection 46 h later. **(B)** _P. berghei_ liver infection quantified by qRT-PCR, 46 h post-sporozoite inoculation into naïve mice (_Pb_—black symbols), mice infected with _T. brucei_ 5 days prior to sporozoite inoculation (_Tb/Pb_—green symbols), or mice that received _T. brucei_ total lysates 30 min, 6 h, 12 h, 24 h and 48 h prior to _P. berghei_ sporozoites inoculation (_TbL/Pb_—blue symbols). Symbols represent the individual values of each mouse in two independent experiments for _T. brucei_ total lysates and one experiment for _P. berghei_ total lysates, and error bars indicate the SEM. **(C)** _P. berghei_ liver infection determined by qRT-PCR, 46 h post-sporozoite inoculation into naïve mice (_Pb_—black symbols), mice infected with _T. brucei_ 5 days prior to sporozoite inoculation (_Tb/Pb_—green symbols), mice that received _T. brucei_ total lysates 30 min prior to _P. berghei_ sporozoites inoculation (_TbL/Pb_—blue symbols), or mice that received _T. brucei_ total lysates 6 h post-_P. berghei_ sporozoite inoculation (_Pb/TbL_—green symbols). Symbols represent the individual values of each mouse in two independent experiments and error bars indicate the SEM. For **(B)** and **(C)** one-way ANOVA with post-test Dunnett was employed to assess the statistical significance of differences between experimental groups. ns, not significant, *$p<0.05$, **$p<0.01$, ***$p<0.001$, ****$p<0.0001$.

mice 6 h after the injection of _P. berghei_ sporozoites (**Fig 2A**). Our results showed that trypanosome lysates inhibit an already established _P. berghei_ liver infection, albeit to an extent lower than that observed when the injection of _T. brucei_ total lysates is performed prior to sporozoite inoculation (**Fig 2C**).

Altogether, our results indicate not only that the protection elicited by *T. brucei* total lysates against *P. berghei* liver infection is time-dependent, but also that the trypanosome lysates are able to inhibit a recently established *P. berghei* liver infection.

## *T. brucei* lysates do not cause major liver damage

It has been previously described that an infection by *T. brucei* results in liver damage and impaired hepatic function [27, 45, 46]. Thus, we hypothesized that these hepatic alterations could be the cause of the decreased number of *P. berghei*-infected hepatocytes following the injection of *T. brucei* total lysates. To evaluate the hepatocellular damage resulting from the administration of *T. brucei* lysates, histopathological analyses of livers collected from both groups of animals at 46 h of *P. berghei* hepatic infection were performed. As expected, the livers of co-infected mice displayed hepatocyte damage and immune cell infiltrates, namely mononuclear inflammatory cells. Conversely, the livers of mice that received trypanosome lysates, similarly to those of mice infected only with *Plasmodium*, displayed minimal tissue damage and no infiltration of immune inflammatory cells (**Fig 3A**).

Although *T. brucei* lysates did not promote microscopically detectable hepatocyte apoptosis, liver function could still be compromised. Thus, we measured several biochemical parameters in naïve mice, mice infected with *T. brucei* or previously injected with *T. brucei* total lysates, at a time-point corresponding to the 46 h of *P. berghei* hepatic infection. In agreement with previous reports, we observed that mice infected by *T. brucei* display increased alanine aminotransferase (ALT), aspartate aminotransferase (AST), bilirubin and decreased glucose levels compared to naïve mice, suggestive of severe liver injury [47–49]. Conversely, mice that received trypanosome lysates did not display altered serum levels for these biochemical markers, relative to naïve mice (**Fig 3B**).

Collectively, our data indicate that trypanosome lysates do not induce hepatocyte damage, nor do they severely compromise liver function, hence suggesting that the reduction in *P. berghei* hepatic load by *T. brucei* infection does not result from lysate-induced liver injury.

## The impairment of *P. berghei* liver infection by *T. brucei* lysates is independent of lymphoid and myeloid immune cells

To assess the mechanism of impairment of *P. berghei* liver infection by *T. brucei* total lysates, we next assessed the possible involvement of the immune response in this process. To test a possible role of lymphocytes, $RAG2^{-/-}\gamma c^{-/-}$ mice, which are genetically deficient for both T, B, natural killer and innate lymphoid cell lymphocytes, were injected with *T. brucei* total lysates prior to sporozoite inoculation. The quantification of *P. berghei* hepatic infection at 46 hpi showed that co-infected and trypanosome lysate-injected $RAG2^{-/-}\gamma c^{-/-}$ mice displayed a reduction of *P. berghei* hepatic infection (~80% and 65%, respectively) similar to that observed in wild-type mice (**Fig 4A**). This result suggests that lymphocytes do not mediate the impairment of *P. berghei* liver infection by *T. brucei* lysates.

It has been previously shown that macrophages are not involved in the impairment of *P. berghei* liver infection by a primary *T. brucei* infection [27]. To investigate whether this is also the case following injection of *T. brucei* total lysates, phagocytic cells from either naïve mice or from mice that received *T. brucei* total lysates were depleted by injection of clodronate-filled liposomes 48 h prior to sporozoite inoculation [50]. The efficiency of depletion was confirmed by the quantification of liver mRNA levels of specific macrophage marker genes (*Clec4f*, *F4/80* and *CD68*) [51] (**S2A Fig**). We found that phagocytic cell-depleted mice that received *T. brucei* total lysates displayed a reduced *P. berghei* liver infection, similar to that observed in their

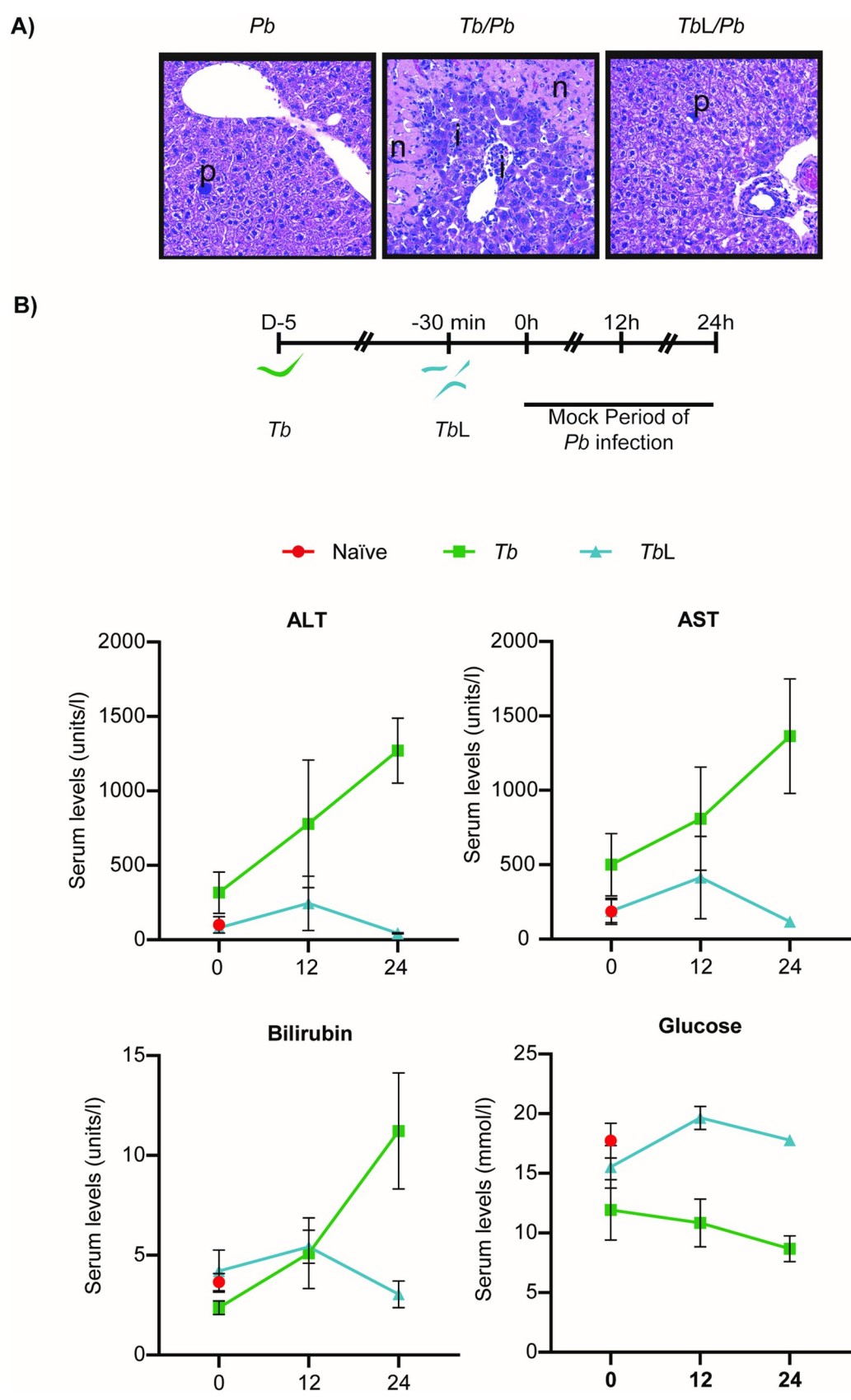

**Fig 3. *T. brucei* lysates do not severely compromise liver function. (A)** Representative microphotographs of liver 46 h after injection of *P. berghei* sporozoites into naïve mice, mice infected 5 days earlier with *T. brucei*, or mice that received *T. brucei* total lysates; depicted are the inflammatory cell infiltrates ("i"), hepatocellular damage/apoptosis ("n") and *P. berghei* EEFs ("p"). Hematoxylin and Eosin. Original magnification 10x. **(B)** Quantification of levels of biochemical parameters in serum. The graphics show the ALT, AST, bilirubin and glucose quantification in serum of naïve mice (Naïve—red dots), mice infected 5 days earlier with *T. brucei* (*Tb*—green lines) or mice that received lysates of trypanosomes (*Tb*L—blue lines). The time points indicated on the XX-axis correspond to the hours of *P. berghei* infection, if sporozoites had been injected. Dots represent the mean values of three to six mice from one independent experiment, with error bars indicating the SEM.

non-treated counterparts (**Fig 4B**), indicating that the impairment of *P. berghei* hepatic infection by *T. brucei* total lysates is independent of macrophages.

Even though the observed transcript levels of *Clec4f* indicated that very few liver-resident macrophages remained in the liver after clodronate treatment, the fact that the mRNA expression of the other marker genes employed was not negligible may suggest monocyte recruitment to the liver. Thus, in order to evaluate whether liver-infiltrated monocytes were responsible for the inhibition of *P. berghei* liver infection following injection of *T. brucei* lysates, monocyte recruitment was blocked by administration of anti-CCR2 antibody 48 and 24 h prior to sporozoite inoculation to either naïve mice, *T. brucei*-infected mice or mice that received *T. brucei* total lysates. The ability of the antibody to prevent monocyte recruitment to the liver was confirmed by flow cytometry analysis, which showed that the number of monocytes in the liver was decreased in all groups of mice, relative to their non-treated counterparts (**S2B Fig**). The quantification of *P. berghei* hepatic load revealed that mice that received the anti-CCR2 antibody and either lysed or live trypanosomes presented a marked impairment of *Plasmodium* liver infection, relative to the respective control mice (**Fig 4C**), indicating that monocytes recruited to the liver are not involved in the phenotype under study.

Redundant or compensatory mechanisms are likely to occur when a subset of cells is depleted [52, 53]. In order to address this issue, mice were immunosuppressed by whole-body lethal γ-irradiation, which markedly decreases the number of circulating leukocytes and mitotically active cells, including bone-marrow hematopoietic cells [54, 55], 24 h prior to sporozoite injection. γ-irradiated mice infected only with *P. berghei* showed a higher *Plasmodium* parasite load than their non-γ-irradiated counterparts (**S2C Fig**), confirming the success of the immunosuppression. We found that γ-irradiated mice that received *T. brucei* total lysates prior to sporozoite inoculation displayed a reduction in *P. berghei* infection load relative to the respective control mice (**Fig 4D**), suggesting that neither lymphoid nor myeloid cells are involved in the trypanosome-mediated inhibitory effect.

We conclude that the impairment of *P. berghei* liver infection caused by *T. brucei* total lysates is independent of lymphoid and myeloid immune cells.

## The impairment of *P. berghei* liver infection is mediated by a protein component of *T. brucei* lysates

A cell lysate is a complex mixture that includes proteins, nucleic acids and lipids, among other molecules. To assess whether trypanosome-derived DNA or proteins were responsible for the reduction of *P. berghei* hepatic infection, lysates treated with either DNAse or proteinase K to degrade their DNA or proteins, respectively, were administered to mice 30 min prior to *P. berghei* sporozoite injection, and the *Plasmodium* liver load was quantified (**Fig 5A**). We found that mice that received DNAse-treated lysates displayed a ~70% reduction in parasite load relative to mice infected only with *P. berghei*, similar to that observed in mice that received whole

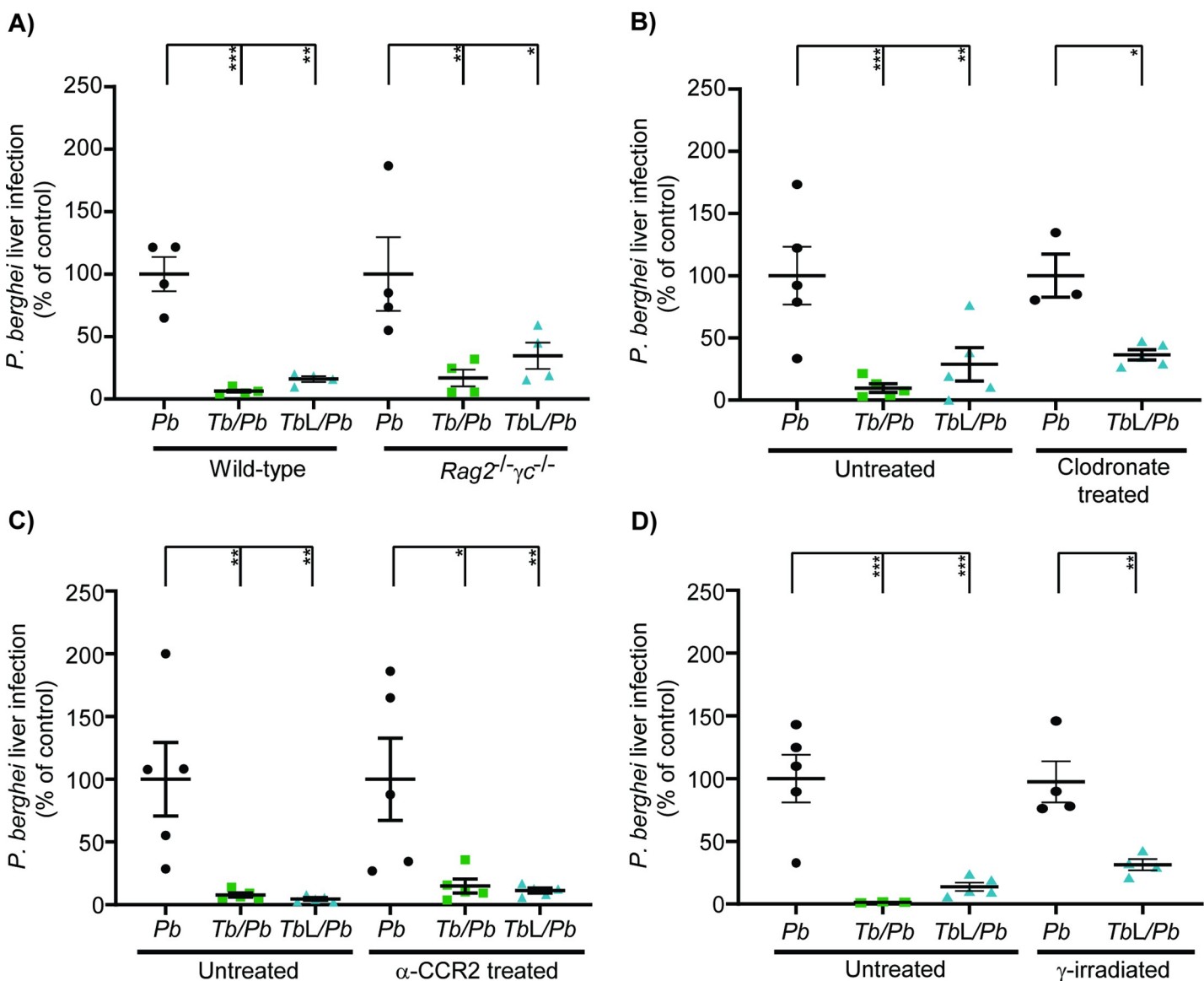

**Fig 4. Impairment of *P. berghei* liver infection by *T. brucei* lysates is independent of lymphoid and myeloid immune cells.** (A) *P. berghei* liver infection quantified by qRT-PCR 46 h after injection of *P. berghei* sporozoites into wild-type and *RAG2⁻/⁻γc⁻/⁻* mice, either naïve (*Pb*—black symbols), infected 5 days earlier with *T. brucei* (*Tb/Pb*—green symbols), or mice that received *T. brucei* total lysates (*TbL/Pb*—blue symbols) 30 min prior to sporozoite inoculation. Symbols represent the individual values of each mouse of one independent experiment and error bars indicate the SEM. **(B)** *P. berghei* liver infection quantified by qRT-PCR 46 h after injection of *P. berghei* sporozoites into naïve mice (*Pb*—black symbols), infected 5 days earlier with *T. brucei* (*Tb/Pb*—green symbols), or mice that received *T. brucei* total lysates (*TbL/Pb*—blue symbols) 30 min prior to sporozoite inoculation, non- or clodronate-treated 48 h prior to *P. berghei* infection. Symbols represent the individual values of each mouse of one independent experiment and error bars indicate the SEM. **(C)** *P. berghei* liver infection quantified by qRT-PCR 46 h after injection of *P. berghei* sporozoites into naïve mice (*Pb*—black symbols), infected 5 days earlier with *T. brucei* (*Tb/Pb*—green symbols), or mice that received *T. brucei* total lysates (*TbL/Pb*—blue symbols) 30 min prior to sporozoite inoculation, administered or not with anti-CCR2. Symbols represent the individual values of each mouse of one independent experiment and error bars indicate the SEM. **(D)** *P. berghei* liver infection quantified by qRT-PCR 46 h after injection of *P. berghei* sporozoites into non- and γ-irradiated mice, either naïve (*Pb*—black symbols), infected 5 days earlier with *T. brucei* (*Tb/Pb*—green symbols), or mice that received *T. brucei* total lysates (*TbL/Pb*—blue symbols) 30 min prior to sporozoite inoculation. Symbols represent the individual values of each mouse of one independent experiment and error bars indicate the SEM. The one-way ANOVA with post-test Dunnett was employed to assess the statistical significance of differences between groups. * $p < 0.05$, ** $p < 0.01$ and *** $p < 0.001$.

lysates. In contrast, *T. brucei* lysates treated with proteinase K did not impair *P. berghei* infection, indicating that the trypanosome lysate component responsible for inhibiting *Plasmodium* infection is a protein.

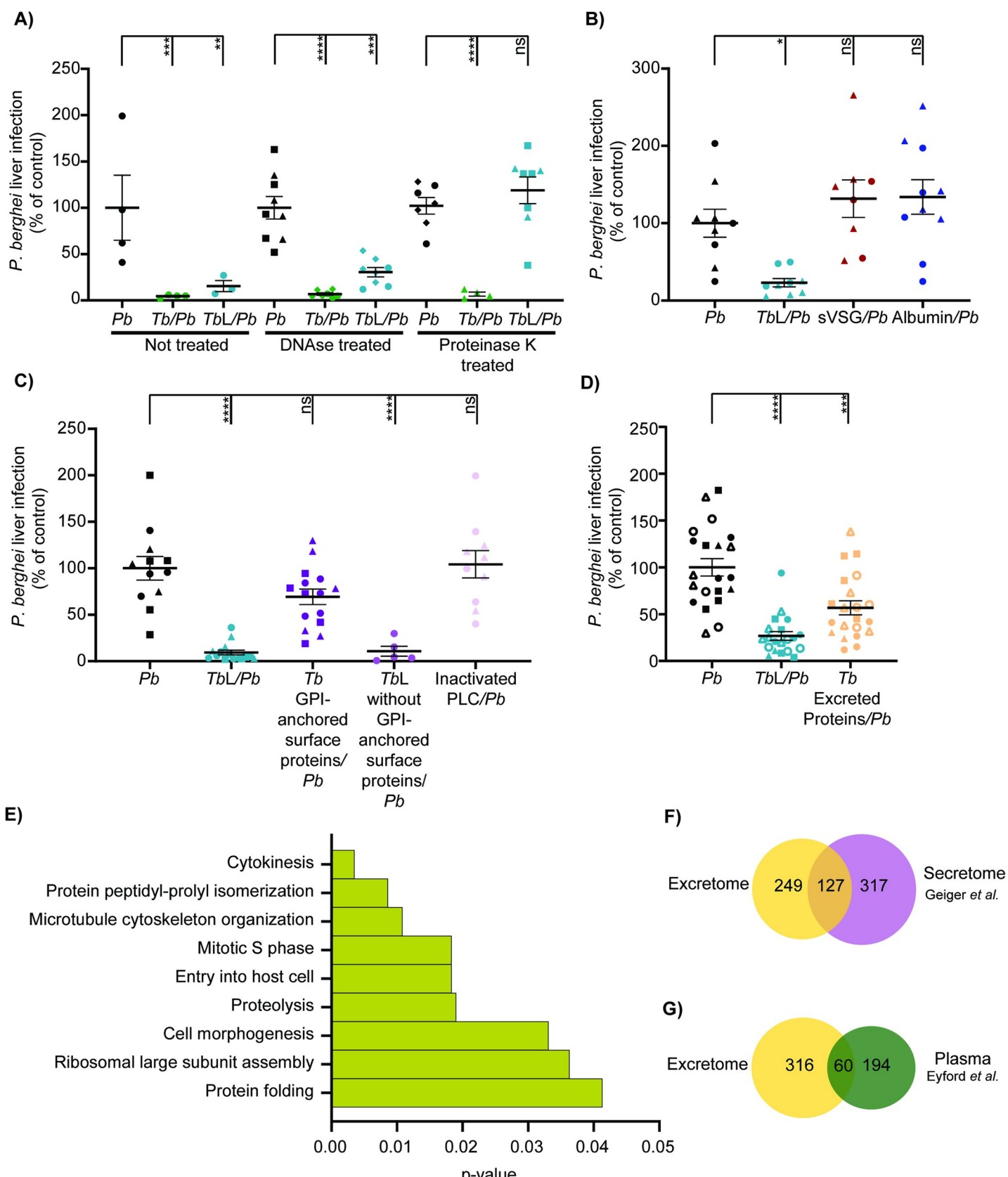

**Fig 5. The impairment of *P. berghei* liver infection by total lysates is mediated by their *T. brucei* protein component.** **(A)** *P. berghei* liver infection load quantified by qRT-PCR 46 h after injection of sporozoites into naïve mice (*Pb*—black symbols), mice infected 5 days earlier with *T. brucei* (*Tb/Pb*—green symbols), or mice that received either the *T. brucei* total lysates (*TbL/Pb*—blue symbols), treated or not with DNAse or proteinase K, or the respective vehicles 30 min prior to sporozoite inoculation. Symbols represent the individual values of each mouse of two independent experiments and error bars indicate the SEM. **(B)** *P. berghei* liver infection load quantified by qRT-PCR 46 h after injection of sporozoites into naïve mice (*Pb*—black symbols), mice that received either *T. brucei* total lysates (*TbL/Pb*—blue symbols), mice that received sVSG (sVSG/*Pb*—red symbols), or mice that received albumin (Albumin/*Pb*—dark blue symbols) 30 min prior to sporozoite inoculation. Symbols represent the individual values of each mouse of two independent experiments and error bars indicate the SEM. **(C)** *P. berghei* liver infection load quantified by qRT-PCR 46 h after injection of sporozoites into naïve mice (*Pb*—black symbols), mice that received *T. brucei* total lysates (*TbL/Pb*—blue symbols), trypanosome GPI-surface proteins (*Tb*GPI-anchored surface proteins/*Pb*—dark purple symbols), *T. brucei* lysates without GPI-surface proteins (*TbL* without GPI-anchored surface proteins/*Pb*—light purple symbols) or the respective vehicle (inactivated PLC/*Pb*—pink symbols) 30 min prior to sporozoite inoculation. Symbols represent the individual values of each mouse of three independent experiments and error bars indicate the SEM. **(D)** *P. berghei* liver infection load quantified by qRT-PCR 46 h after injection of sporozoites into naïve mice (*Pb*—black symbols), mice that received *T. brucei* total lysates (*TbL/Pb*—blue symbols), trypanosome excreted proteins (*Tb*Excreted Proteins/*Pb*—yellow symbols) 30 min prior to sporozoite inoculation. Symbols represent the individual values of each mouse of five independent experiments and error bars indicate the SEM. **(E)** Enriched GO terms among excreted *T. brucei* proteins. Biological process category included. **(F)** Venn diagram of the comparative analysis between *T. brucei* published secretome (30) (purple) and the excreted proteins identified in this study (yellow). **(G)** Venn diagram of the comparative analysis between *T. brucei* proteins identified in plasma form *T. b. rhodesiense* infected patients (29) (green) and excreted proteins identified in this study (yellow). For **(A)** to **(D)** The one-way ANOVA with post-test Dunnett was employed to assess the statistical significance of differences between groups. ns, not significant, $^*p < 0.05$, $^{**}p < 0.01$, $^{***}p < 0.001$ and $^{****}p < 0.0001$.

VSG is the most abundant protein in the bloodstream forms of *T. brucei*, comprising ~10% of total protein content, and it is anchored to the plasma membrane through a glycosylphosphatidylinositol (GPI) [56–59]. Since VSG has been previously found in the parasite's excretome [30], we investigated whether this protein was responsible for the observed inhibitory effect on *P. berghei* hepatic infection. To this end, the soluble form of VSG (sVSG), which is also found in the serum of infected animals, and which consists of a polypeptide and a glycosyl-inositol-phosphate (GIP) moiety [60], was purified (**S3 Fig**), and an amount of this protein equivalent to that present in 1 x 10$^8$ trypanosomes was injected into mice prior to sporozoite inoculation. Quantification of the *Plasmodium* liver load showed that the administration of sVSG did not reduce *P. berghei* liver infection relative to that of control mice (**Fig 5B**), suggesting that VSG is not involved in the phenotype under investigation. As expected, injection of a similar amount of albumin also did not cause an inhibitory effect on *P. berghei* hepatic infection.

Next, we investigated whether other cell-surface glycoproteins could be responsible for the observed inhibitory effect. Trypanosomes were treated with an exogenous phospholipase C (PLC), which cleaves the GPI-anchors immediately before the phosphate group, releasing the polypeptides and glycan moiety into a soluble fraction, while the dimyristoylglycerol remains attached to the parasite surface [60]. The soluble fraction was then injected into mice, followed by injection of sporozoites. As expected, we observed that the parasite liver load of mice that received the GPI-anchored surface proteins fraction was not different from that of control mice. Additionally, we observed that the *T. brucei* lysates without the GPI-anchored surface proteins retained their inhibitory effect, which was not impacted by PLC's enzymatic activity (**Fig 5C**). Altogether, these results indicate that the polypeptides of GPI-anchored proteins are not involved in the inhibitory phenotype.

We subsequently tested whether the *T. brucei* protein(s) responsible for the impairment of *Plasmodium* liver infection is/were present in more than one stage of the trypanosome life cycle. To investigate this, total lysates of either *T. brucei* bloodstream forms (BSF) or procyclic forms (PCF, found in the insect vector) were administered to mice prior to sporozoite injection, and *Plasmodium* hepatic load was assessed. Interestingly, we found that the lysates of both trypanosome life cycle stages reduced *P. berghei* hepatic infection, relative to that of mice infected only with *Plasmodium* (**S4 Fig**). This result suggests that the protein(s) that impair liver infection by *P. berghei* is/are either expressed in more than one stage of the trypanosome

life cycle, or that a different set of proteins has a similar *Plasmodium* inhibitory activity in both stages of the trypanosome life cycle.

To identify the inhibitory trypanosome protein(s), we started by assessing whether they were excreted. To assess this, trypanosomes were incubated in PBS for 1 h at room temperature and the proteins excreted during this period were collected. The fraction containing the excreted proteins was subsequently injected into mice prior to sporozoite inoculation and *P. berghei* liver infection was quantified (**Fig 5D**). We found that the excreted protein fraction reduces *P. berghei* liver infection to ~40% of the liver load observed mice infected only with *P. berghei*, indicating that the impairment of *P. berghei* liver infection is mediated by one or more excreted *T. brucei* proteins.

Extracellular vesicles are used by several microorganisms as vehicles for delivery of proteins, lipids and nucleic acids [61]. It has been shown that *T. brucei* also releases EVs that fuse with the host erythrocyte membrane, altering the physical properties of the membrane, and causing the RBC to be cleared from circulation [43]. Since the excreted protein fraction of *T. brucei* reduces *P. berghei* liver infection, we hypothesized that *T. brucei* EVs might be involved in this process. To assess this, we purified *T. brucei* EVs and used transmission electron microscopy (TEM) to assess their shape and size ($\approx$150 nm) and confirm their identity (**S5A Fig**). EVs were injected into mice prior to sporozoite inoculation and *Plasmodium* liver infection was assessed as above. Mice that received *T. brucei* EVs did not present an impairment of *Plasmodium* liver infection, relative to the respective control mice, which indicates that *T. brucei* EVs are not involved in the phenotype under study (**S5B Fig**).Mass spectrometry analysis of the active excreted fraction (which we call here, excretome) revealed it to be composed by 376 proteins (**S1 Table**). Gene ontology (GO) analysis showed that most of the proteins belong to biological processes of interaction with the host, protein folding and translation, cell cycle and cellular organization (**Figs 5E and S6**). Having previously excluded proteins with a GPI-anchor, we used a GPI-anchor predictor [62] to disregard 13 predicted GPI-anchored proteins (**S1 Table**), from the list of candidate proteins responsible for the impairment of *P. berghei* liver infection. A comparative analysis between our 376 excreted proteins and the published *T. brucei* secretome [30] indicated that 127 proteins are common between the two data sets (**Fig 5F**). This difference was not surprising since the secretion/excretion treatments were different between the two studies. Interestingly, 60 proteins (**S2 Table**) found in the excretome have been previously identified in the plasma of *T. b. rhodesiense* infected patients [29] (**Fig 5G**). Thus, it is possible that one of these 60 proteins could be responsible for the observed inhibitory activity.

Collectively, our data show that trypanosome BSF excreted protein(s) impair(s) *P. berghei* hepatic infection. Importantly, VSG, the most abundant surface BSF trypanosome protein, does not mediate the impairment of *P. berghei* liver infection.

## Discussion

Considering that several microorganisms share a geographic distribution with *Plasmodium*, co-infections between those and the latter are likely to occur [19–25]. Some reports state that the course of the *Plasmodium* infection can be aggravated [63] or impaired [28] by the presence of a second pathogen. Due to the incidence and geographical distribution of *Plasmodium* and *T. brucei*, malaria is commonly found in patients with HAT [26]. In a previous report, we showed that an ongoing *T. brucei* infection was able to inhibit a secondary *P. berghei* liver infection in mice [27]. It is of the utmost importance to understand the mechanism behind this phenomenon, since it can not only lead to the identification of a novel mechanism at play

during a co-infection between two different parasite species, but it may also unveil novel strategies of malaria control.

In this work, we showed that the anti-*Plasmodium* activity exhibited by *T. brucei* does not require live trypanosomes. Serum transfer from a *T. brucei*-infected mouse into a naïve animal is able to protect the latter one from a subsequent *P. berghei* hepatic infection. By injecting *T. brucei* total lysates into mice prior to or after *P. berghei* sporozoite inoculation, we further showed that the serum molecules conferring protection against *Plasmodium* liver infection were trypanosome-derived. Of note, these molecules protected mice from the development of severe pathology, increasing host survival, which may be explained by the delay in the appearance of *Plasmodium* parasitaemia [64, 65]. These results suggest that a patient co-infected with *Plasmodium* and *T. brucei*, even with undetectable trypanosome parasitaemia, might display some level of protection against the malaria parasite, as long as the trypanosome molecules are in circulation.

Liver injury can be a result of the immune response to the infectious agent and may involve the recruitment of immune cells to the liver, which leads to hepatic inflammation and tissue damage [66]. In fact, liver damage [46, 67] and mononuclear cell infiltrates [35, 68] have been observed during *T. brucei* infection [27]. Consistently, our histopathological analysis showed that the livers of *T. brucei*-infected mice presented signs of liver damage and compromised hepatic function. However, these signs were absent from the livers of mice injected with *T. brucei* lysates. Likewise, and as expected [47–49], increased levels of ALT, AST and bilirubin, and decreased levels of glucose were observed in *T. brucei*-infected mice but not in animals that received *T. brucei* total lysates. Additionally, and contrarily to what was shown for the co-infection between *T. brucei* and *P. berghei* [27], the impairment of *P. berghei* hepatic infection by *T. brucei* total lysates appears to be independent of liver damage and of the host immune response.

Additionally, we found that the impairment of *P. berghei* hepatic infection by total *T. brucei* lysates is mediated by a protein or an element of a protein, such as the oligosaccharides present on surface glycoproteins, likely present in both bloodstream and procyclic parasite forms. Our results suggest that these proteins are excreted by trypanosomes into the bloodstream of the host. Knowing that a primary *T. brucei* infection protects mice against malaria [27] and that passive transfer of serum from *T. brucei*-infected animals into naïve mice led to an impairment of *P. berghei* hepatic infection, we propose that during co-infection between *P. berghei* and *T. brucei*, the latter excretes a protein that is relatively stable and that will subsequently inhibit hepatic infection by the malaria parasite.

While *T. brucei* lysates injected prior to sporozoite inoculation strongly impair the *P. berghei* liver infection, suggestive of a strong effect on the hepatocyte, when these lysates are injected after the infection is established, a marked reduction in *P. berghei* liver load is also observed. One may therefore speculate that the *T. brucei* excreted proteins may be impairing *P. berghei* liver infection prior and post-invasion. Trypanosomes may be rendering the hepatocytes more sensitive to invasion by pathogens and immediately activate programmed cell death pathways, which have been described as defense mechanisms against intracellular bacterial mono-infections [69]. Alternatively, trypanosome molecules may bind or fuse with the hepatocyte membrane. In that event, when the sporozoite productively invades the hepatocyte, the parasitophorous vacuole membrane may bear at its surface not only host proteins, but may be also decorated with trypanosome molecules, no longer effectively protecting *Plasmodium* from elimination. It has also been reported that *P. berghei* parasites can be eliminated by a lysosome-mediated mechanism that is independent of the immune system [70]. It is possible that *T. brucei* proteins may render *Plasmodium* intra-hepatic forms more sensitive to lysosomes, and/or may trigger a signaling cascade that induces the recruitment of the lysosomes to the

parasite's parasitophorous vacuole membrane, leading to an acidification of the vacuole and subsequent elimination of the intra-hepatic *Plasmodium* forms [70].

From the 376 excreted proteins detected by spectrometry analysis, we excluded 13 predicted GPI-anchored proteins from being responsible for the anti-*Plasmodium* activity. Interestingly, our data revealed the presence of a great diversity of excreted proteins (**S1 Table**), with a wide variety of functions. This is in agreement with proteomic studies of proteins secreted not only by *T. brucei* [30], but also by several other organisms, like *P. berghei* [71], *Leishmania donovani* [72] and *Trypanosoma cruzi* [73], that share similarities with the excretome presented in this study. At this stage, the identity of the protein(s) responsible for the inhibitory activity remain(s) unknown. Multifunctional proteins, known as moonlight proteins, have been described in multiple organisms [74, 75]. Several moonlight proteins have been characterized in protozoan parasites and shown to play different roles on the parasite and on the host. For instance, although the primary function of *Leishmania donovani*'s elongation factor-1α is to mediate translation elongation, this protein was also described to play an important role on macrophage deactivation [74]; the *P. falciparum* enzyme aldolase, commonly known for its role in glycolysis [76], has also been shown to be involved in host-cell invasion [75]. Of note, *T. brucei* has only one moonlight protein described in the moonlight proteins database [77]. Interestingly, proteins from the excretome have been described as moonlight proteins in other organisms, for example: *P. falciparum*'s and *Toxoplasma gondii*'s aldolase, *Leishmania donovani*'s elongation factor-1α, *Streptococcus agalactiae*'s phosphoglycerate kinase, *Lactococcus lactis*'s 6-phosphofructokinase. Thus, it can be speculated that one or more moonlight proteins might be involved in the impairment of *P. berghei* hepatic infection by *T. brucei*.

The fact that the serum transfer of *T. brucei* infected mouse into a naïve animal inhibits a subsequent *P. berghei* hepatic infection suggests that the excreted trypanosome protein has marked stability in the serum. Indeed, our results showed that the lysates lose the protective effect with time and that when sporozoites are inoculated after 12 h of lysate injection, the inhibitory effect is no longer statistically significant. In the serum, proteins have different half-lives. For example, while the half-life of IgG in mouse serum is 95 h, albumin has a half-life of 35 h [78] and the ornithine decarboxylase has a half-life that can go from 5 to 30 minutes [79]. However, it has also been shown that the lifespan of a protein can differ between tissues, as protein turnover in the brain is slower (average half-life of 9 days) than in the blood (3.5 days) or liver (3 days) [80]. The protective effect of *T. brucei* lysates may decrease not only due the kinetics of protein clearance in mice, but also in accordance to the half-life of the specific protein(s) in question. From the 60 proteins that are common between the excretome identified in this study and those found in the plasma of *T. b. rhodesiense* infected-patients [29], we speculate that at least one protein might be responsible for the observed inhibitory activity. Future studies will aim at identifying the trypanosome molecules responsible for the inhibition of *Plasmodium* liver infection.

The present study reinforces the importance of understanding the impact of co-infections, since it may have important repercussions on human and animal health. Overall, our results unveiled that during a *T. brucei* infection, parasitic excreted proteins are likely released into and circulate in the bloodstream of the infected host, protecting it against a subsequent hepatic infection by *Plasmodium* parasites. Future unraveling of the mechanism behind this phenotype may contribute to a deeper understanding of *Plasmodium* biology, and possibly to the development of novel, urgently needed, antiplasmodial intervention strategies.

## Supporting information

**S1 Fig. *T. brucei* total lysates impair a hepatic infection by *P. yoelii*.** *P. berghei* (*Pb*) and *P. yoelii* (*Py*) liver infection load quantified by qRT-PCR 46 h after injection of *P. berghei*

sporozoites into naïve mice (*Pb*–black symbols; *Py*–orange symbols), mice infected 5 days earlier with *T. brucei* (*Tb/Pb*–green symbols or *Tb/Py*–yellow symbols), or mice that received lysates of trypanosomes (*Tb*L/*Pb*–blue symbols; or *Tb*L/*Py*–dark orange symbols) 30 min prior to sporozoite inoculation. Symbols represent the individual values of each mouse in one independent experiment and error bars indicate the SEM. The one-way ANOVA with post-test Dunnett was employed to assess the statistical significance of differences between groups. ns, not significant, **p<0.01 and ****p<0.0001.
(TIF)

**S2 Fig. Confirmation of macrophage depletion and prevention of monocyte recruitment to the liver. (A)** *Clec4f*, *F4/80* and *CD68* gene expression quantification by qRT-PCR in the liver 46 h after injection of *P. berghei* sporozoites into naïve mice (*Pb*–black bars), or mice that received lysates of trypanosomes (*Tb*L/*Pb*—blue bars) 30 min prior to sporozoite inoculation, non- or clodronate-treated 48 h prior to *P. berghei* infection. Bars represent the mean values of one independent experiment and error bars indicate the SEM. **(B)** Assessment of monocyte abundance in the liver by flow cytometry in the liver 46 h after injection of *P. berghei* sporozoites into naïve mice (*Pb*—black bars), infected 5 days earlier with *T. brucei* (*Tb/Pb*—green bars), or mice that received lysates of trypanosomes (*Tb*L/*Pb*—blue bars) 30 min prior to sporozoite inoculation, administered or not with anti-CCR2. Bars represent the mean values of one independent experiment and error bars indicate the SEM. **(C)** *P. berghei* liver infection quantified by qRT-PCR 46 h after injection of *P. berghei* sporozoites into non- and γ-irradiated naïve mice, without normalization. Symbols represent the individual values of each mouse of one independent experiment and error bars indicate the SEM. For **(A)** and **(B)** The Mann-Whitney test was employed to assess the statistical significance of differences between experimental groups. ns, not significant, *p<0.05 and **p<0.01. For **(C)** an Unpaired t test was employed to assess the statistical significance of differences between experimental groups. ns, not significant.
(TIF)

**S3 Fig. Confirmation of sVSG purification by SDS-PAGE, stained with Coomassie.** After purification of VSG2, 20 μL of purified sample were loaded on a 10% SDS-PAGE gel. The predicted molecular weight of VSG2 is ≈51 kDa.
(TIF)

**S4 Fig. *T. brucei* total lysates of procyclic forms are able to impair a secondary *P. berghei* liver infection.** *P. berghei* liver infection load quantified by qRT-PCR 6h and 46 h after injection of *P. berghei* sporozoites into naïve mice (*Pb*–black symbols), mice infected 5 days earlier with *T. brucei* (*Tb/Pb*–green symbols), or mice that received lysates of either bloodstream form trypanosomes (BSF *Tb*L/*Pb*—blue bars) or procyclic form trypanosomes (PCF *Tb*L/*Pb*—yellow symbols) 30 min prior to sporozoite inoculation. Symbols represent the individual values of each mouse in three independent experiments, in which each symbol format corresponds to an individual experiment, and error bars indicate the SEM. The one-way ANOVA with post-test Dunnett was employed to assess the statistical significance of differences between groups. ns, not significant and ****p<0.0001.
(TIF)

**S5 Fig. *T. brucei* extracellular vesicles do not impair a secondary *P. berghei* liver infection. (A)** TEM of purified EVs, observed by negative staining. **(B)** *P. berghei* liver infection load quantified by qRT-PCR 46 h after injection of sporozoites into naïve mice (*Pb*—black symbols), mice that received *T. brucei* total lysates (*Tb*L/*Pb*—blue symbols), trypanosome EVs (EVs/*Pb*—orange symbols) 30 min prior to sporozoite inoculation. Symbols represent the

individual values of each mouse of three independent experiments and error bars indicate the SEM. The one-way ANOVA with post-test Dunnett was employed to assess the statistical significance of differences between groups. ns, not significant and $^{**}$p<0.01.
(TIF)

**S6 Fig. Enriched GO terms among detected proteins.** GO categories included: molecular function and cellular component.
(TIF)

**S1 Table. Identified proteins in excretome sample.**
(XLSX)

**S2 Table. Proteins in common between our excretome and proteins identified in plasma from *T. b. rhodesiense* infected patients.**
(XLSX)

## Acknowledgments

We acknowledge the Electron Microscopy Facility at the Instituto Gulbenkian de Ciência for sample processing and imaging. We are very grateful to Bruno Silva-Santos for kindly providing $RAG2^{-/-}\gamma c^{-/-}$; to Patrícia Meireles for the help in experiments involving mice γ-irradiation; to Sara Silva Pereira for the help with the GO term analysis; to Filipa Teixeira for producing *P. berghei*-infected mosquitoes; to Leonor Pinho for managing the production of large amounts of medium; to Silvia A. Synowsky & Sally L. Shirran for running the mass spec samples at BSRC Mass Spectrometry and Proteomics Facility, University of St Andrews and to the Bioimaging, Flow Cytometry, Histology and Comparative Pathology and Rodent facilities of iMM JLA for technical support.

## Author Contributions

**Conceptualization:** Adriana Temporão, Margarida Sanches-Vaz, Miguel Prudêncio, Luisa M. Figueiredo.

**Data curation:** Adriana Temporão, Margarida Sanches-Vaz, Helena Nunes-Cabaço, Terry K. Smith.

**Formal analysis:** Adriana Temporão, Margarida Sanches-Vaz.

**Funding acquisition:** Miguel Prudêncio, Luisa M. Figueiredo.

**Investigation:** Adriana Temporão, Margarida Sanches-Vaz, Rafael Luís, Helena Nunes-Cabaço, Terry K. Smith.

**Methodology:** Adriana Temporão, Margarida Sanches-Vaz, Rafael Luís, Helena Nunes-Cabaço, Terry K. Smith.

**Project administration:** Miguel Prudêncio, Luisa M. Figueiredo.

**Resources:** Miguel Prudêncio, Luisa M. Figueiredo.

**Supervision:** Miguel Prudêncio, Luisa M. Figueiredo.

**Validation:** Adriana Temporão, Margarida Sanches-Vaz, Rafael Luís, Helena Nunes-Cabaço, Terry K. Smith.

**Visualization:** Adriana Temporão.

**Writing – original draft:** Adriana Temporão, Margarida Sanches-Vaz.

**Writing – review & editing:** Miguel Prudêncio, Luisa M. Figueiredo.

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
