## [Decision Letter · Decision Letter 0]

5 Jul 2021

Dear Dr. Figueiredo,

Thank you very much for submitting your manuscript "Excreted Trypanosoma brucei proteins inhibit Plasmodium hepatic infection" for consideration at PLOS Neglected Tropical Diseases. As with all papers reviewed by the journal, your manuscript was reviewed by members of the editorial board and by several independent reviewers. In light of the reviews (below this email), we would like to invite the resubmission of a significantly-revised version that takes into account the reviewers' comments. 

Although both reviewers were very positive about the quality and presentation of the data, both also raised concerns about the lack of clarity pertaining to the mechanism underpinning the observed suppression of infectivity. Specifically, they had questions about whether this impairment is a function of sporozoites or hepatocytes and felt that this would have to be clarified in order to merit publication. Each has suggested an experimental approach to address this question namely, testing serum from T. brucei-infected mice to see if they recognise sporozoites and elucidating whether T. brucei-derived EVs might be fusing to the hepatocytes.

We cannot make any decision about publication until we have seen the revised manuscript and your response to the reviewers' comments. Your revised manuscript is also likely to be sent to reviewers for further evaluation.

Sincerely,

Katerina Artavanis-Tsakonas

Associate Editor

Jan Van Den Abbeele

Deputy Editor

Reviewer's Responses to Questions

**Key Review Criteria Required for Acceptance?**

**Methods**

-Are the objectives of the study clearly articulated with a clear testable hypothesis stated?

-Is the study design appropriate to address the stated objectives?

-Is the population clearly described and appropriate for the hypothesis being tested?

-Is the sample size sufficient to ensure adequate power to address the hypothesis being tested?

-Were correct statistical analysis used to support conclusions?

-Are there concerns about ethical or regulatory requirements being met?

Reviewer #1: This clearly defined and well written study describes the inhibition of Plasmodium berghei liver infection following passive transfer from mice previously infected by T. brucei, and with total protein lysates from T. brucei. In depth and logical biochemical analysis demonstrates that protective efficacy in independent of VSG levels, and that GPI aochored proteins are not responsible for mediating protection. Initial proteomic analysis was then performed to identify the inhibitory trypanosomal components, with mass spec identifying 376 proteins present to potentially contribute to protection, 127 which have been previously defined in the T. brucei secretome. 

This work builds largely on previously work described in Sanches-Vas et al., 2019, and Eyford et al., 2013 – and is a logical and meaningful extension to these findings. Experiments are well designed and described, with appropriate controls. Conclusions are supported by the results, and referencing is correct throughout. Samples sizes, ethics and stats are all correct.

Reviewer #2: Methods are fine - if not exhaustive.

**Results**

-Does the analysis presented match the analysis plan?

-Are the results clearly and completely presented?

-Are the figures (Tables, Images) of sufficient quality for clarity?

Reviewer #1: Analysis is correct, and results are clearly and completely presented. All figures are apporpriate, with one suggested edit (below).

Reviewer #2: Also okay (see general comments below)

**Conclusions**

-Are the conclusions supported by the data presented?

-Are the limitations of analysis clearly described?

-Do the authors discuss how these data can be helpful to advance our understanding of the topic under study?

-Is public health relevance addressed?

Reviewer #1: Conclusions are supported by data - I would like a brief expansion on public health relevance, but this is a minor point.

Reviewer #2: Not sure these are justified. See comments below.

**Editorial and Data Presentation Modifications?**

Reviewer #1: Figure 1G – would it be possible to make the Tb/Pb line a different colour? Very easy to miss at the moment.

Reviewer #2: (No Response)

**Summary and General Comments**

Reviewer #1: This clearly defined and well written study describes the inhibition of Plasmodium berghei liver infection following passive transfer from mice previously infected by T. brucei, and with total protein lysates from T. brucei. In depth and logical biochemical analysis demonstrates that protective efficacy in independent of VSG levels, and that GPI aochored proteins are not responsible for mediating protection. Initial proteomic analysis was then performed to identify the inhibitory trypanosomal components, with mass spec identifying 376 proteins present to potentially contribute to protection, 127 which have been previously defined in the T. brucei secretome. 

This work builds largely on previously work described in Sanches-Vas et al., 2019, and Eyford et al., 2013 – and is a logical and meaningful extension to these findings. Experiments are well designed and described, with appropriate controls. Conclusions are supported by the results, and referencing is correct throughout. Samples sizes, ethics and stats are all correct. 

I have a few minor edits to suggest:

1). Line 133 refers to infection by retro-orbital route – why is this chosen as an infection route? i.v is not entirely representative of natural infection (as the authors state here – 5 infectious bites is standard and easily performed)? 

2). Lines 375-387. A very nice set of experiments to demonstrate that impairment of infection is due to a decrease in the number of infected hepatocytes rather than impairment of intrahepatic replication. This leads to the obvious follow on question – is impairment due to effects on the sporozoite (prior to invasion), or the hepatocyte (post-invasion)? Clarity on this would be welcome – e.g. a IFA against sporozoites would at least demonstrate if serum recognises the surface of the sporozoite pre-infection.

3). Figure 1G – would it be possible to make the Tb/Pb line a different colour? Very easy to miss at the moment. 

4). Lines 454-456 – you conclusively show that protection is time dependent – how does the data you see compare to the natural clearance rate of other proteins in mice after passive transfer (e.g. – of transfused IgG)? Does the protective effect decrease due to the standard kinetics of protein clearance in mice?

Reviewer #2: This paper is a followup to the work of M. Vas, M. Prudencio and L. Figueiredo ("Trypanosoma brucei infection protects mice against malaria", PNAS 2019). Here, the authors drill down to try and identify the substance, released by T.brucei, that blocks infection by plasmodiumm. Toward this, they show that protection from hepatocyte infection can be conferred by:

- passive transfer of serum from T.brucei infected animals

- injection with T.brucei lysates (in a dose dependent manner and without overt toxicity) and only if administered closely enough to inoculation with plasmodium. This very short time window already suggests that protection is not dependent on the development of an immune response (which they also then formally demonstrate in Fig4)but rather dependent on a substance secreted or excreted by the parasite. 

 The mechanism by which this suppression of infectivity takes place is entirely unclear.I strongly suspect that one of the alternative possibilities offered in the discussion (that trypanosome fragments enter and decorate the surface of the hepatocyte thus somehow altering its properties and rendering it more resistant to plasmodium infection) is in fact correct. This is the possibility that material found in serum and in lysates - but which is particulate (like EVs secreted by Tbrucei - decorated with VSG but wholly different than soluble VSG which the authors also test) could in fact incorporate into hepatocyte membranes when injected (as was already shown for RBC membranes by Szempruch et al (PMID: 26771494). This is an experiment that's straightforward to do and could solve the mystery (the experiment would be: generate EVs,e.g. from VSG2 expressing Tbrucei, ideally in PLC-/- background; see if they incorporate using anti-VSG2 antibody to chase the signal; then go backward and search for EVs in lysates and serum of infected animals).

PLOS authors have the option to publish the peer review history of their article (what does this mean?). If published, this will include your full peer review and any attached files.

Reviewer #1: No

Reviewer #2: Yes: Nina Papavasiliou
---

## [Decision Letter · Decision Letter 1]

15 Oct 2021

Dear Dr. Figueiredo,

We are pleased to inform you that your manuscript 'Excreted Trypanosoma brucei proteins inhibit Plasmodium hepatic infection' has been provisionally accepted for publication in PLOS Neglected Tropical Diseases.

Best regards,

Katerina Artavanis-Tsakonas

Associate Editor

Jan Van Den Abbeele

Deputy Editor

Reviewer's Responses to Questions

**Key Review Criteria Required for Acceptance?**

**Methods**

-Are the objectives of the study clearly articulated with a clear testable hypothesis stated?

-Is the study design appropriate to address the stated objectives?

-Is the population clearly described and appropriate for the hypothesis being tested?

-Is the sample size sufficient to ensure adequate power to address the hypothesis being tested?

-Were correct statistical analysis used to support conclusions?

-Are there concerns about ethical or regulatory requirements being met?

Reviewer #2: no concerns

**Results**

-Does the analysis presented match the analysis plan?

-Are the results clearly and completely presented?

-Are the figures (Tables, Images) of sufficient quality for clarity?

Reviewer #2: The results are better clarified and the authors were very responsive to reviewer comments.

**Conclusions**

-Are the conclusions supported by the data presented?

-Are the limitations of analysis clearly described?

-Do the authors discuss how these data can be helpful to advance our understanding of the topic under study?

-Is public health relevance addressed?

Reviewer #2: I'm still not convinced about EVs - looking at Fig S5 it seems that the lack of significance might lie in the 2-3 outliers in the EV experiment. Additionally, it is hard to tell "how many" EVs were injected (only that they were purified from 10^8 cells, but the abundance here might be important). Having said that, I appreciate that the authors at least tried the experiment which, even if not 100% conclusive would point to other possibilities (perhaps the sugars released by soluble proteins can bind glycan receptors on hepatocytes? who knows - that's an entirely different line of experimentation).

**Editorial and Data Presentation Modifications?**

Reviewer #2: Accept.

**Summary and General Comments**

Reviewer #2: See above.

PLOS authors have the option to publish the peer review history of their article (what does this mean?). If published, this will include your full peer review and any attached files.

Reviewer #2: **Yes: **Nina Papavasiliou

---

## [Editor Report · Acceptance letter]

26 Oct 2021

Dear Dr. Figueiredo,

We are delighted to inform you that your manuscript, "Excreted Trypanosoma brucei proteins inhibit Plasmodium hepatic infection," has been formally accepted for publication in PLOS Neglected Tropical Diseases.

Best regards,

Shaden Kamhawi

co-Editor-in-Chief

Paul Brindley

co-Editor-in-Chief
